# The Effects of the Schroth Method on the Cobb Angle, Angle of Trunk Rotation, Pulmonary Function, and Health-Related Quality of Life in Adolescent Idiopathic Scoliosis: A Narrative Review

**DOI:** 10.3390/healthcare13202631

**Published:** 2025-10-20

**Authors:** Ana Belén Jiménez-Jiménez, Elena Gámez-Centeno, Javier Muñoz-Paz, María Nieves Muñoz-Alcaraz, Fernando Jesús Mayordomo-Riera

**Affiliations:** 1Interlevel Clinical Management Unit of Physical Medicine and Rehabilitation, Reina Sofia University Hospital, Córdoba and Guadalquivir Health District, 14011 Córdoba, Spain; anab.jimenez.jimenez.sspa@juntadeandalucia.es (A.B.J.-J.); javier.munoz.paz.sspa@juntadeandalucia.es (J.M.-P.); fernandoj.mayordomo.sspa@juntadeandalucia.es (F.J.M.-R.); 2Maimónides Biomedical Research Institute of Cordoba (IMIBIC), Reina Sofia University Hospital, University of Córdoba, 14004 Córdoba, Spain; 3Department of de Applied Physics, Radiology and Physical Medicine, University of Córdoba, 14004 Córdoba, Spain; 4Faculty of Medicine and Nursing, University of Córdoba, 14004 Córdoba, Spain; h92gacee@uco.es

**Keywords:** adolescent idiopathic scoliosis, Schroth method, Cobb angle, angle of trunk rotation, pulmonary function, health-related quality of life

## Abstract

**Background/Objectives**: Adolescent idiopathic scoliosis (AIS) is a three-dimensional deformity of the spine that can negatively impact on quality of life, pulmonary function, and body image. Its conservative management includes various interventions, among which the Schroth method stands out. This approach is based on three-dimensional corrective exercises and rotational breathing. This review aimed to analyze the effectiveness of the Schroth method, applied either alone or in combination with other conservative therapies, on variables such as Cobb angle, angle of trunk rotation (ATR), pulmonary function, and health-related quality of life in patients with AIS. **Methods**: A scientific literature search was conducted using the PubMed database. We searched for randomized controlling trials (RCTs), systematic reviews, and meta-analyses reported in English from 2020 to 2025. Different combinations of the terms and MeSH terms “adolescent”, “idiopathic”, “scoliosis”, and “Schroth” connected with various Boolean operators. **Results**: Overall, 82 articles were reviewed from the selected database. After removing duplicated papers and title/abstract screening, 13 studies were included in our review. The results showed that the Schroth method proved effective in reducing the Cobb angle and ATR, particularly in patients with mild curves and in early stages of skeletal growth. Improvements were also observed in health-related quality of life and aesthetic perception, and to a lesser extent, in pulmonary function. Moreover, therapeutic adherence and treatment continuity were important to maintaining long-term benefits. **Conclusions**: The Schroth method could be an effective treatment associated with orthopedic treatment, yielding satisfactory results. Its implementation requires structured programs, professional supervision, and strategies to enhance therapeutic adherence. Nevertheless, to validate its long-term effectiveness, we need more homogeneous studies with longer follow-up durations.

## 1. Introduction

Adolescent idiopathic scoliosis (AIS) is a three-dimensional deformity of the spine that appears in young people between the ages of 10 and 18 years. The definition of this clinical entity is determined by the appearance of a curvature in the frontal plane of 10 degrees or more in the Cobb angle, a vertebral rotation in the axial plane, and alteration of the physiological sagittal curvature [1]. Its etiology is still unknown; it is not associated with certain diseases such as congenital malformations, syndromic conditions, or neuromuscular diseases, among others. However, some articles indicate that it is a polygenic disorder linked to genetic factors related to the prognosis of AIS [1,2]. Its prevalence is around 1% to 4% of adolescents, representing the most frequent form. Typically, it manifests in healthy individuals during the pubertal development stage, and it affects females at a 10:1 ratio, especially in curves with a Cobb angle greater than 40 degrees [3].

Classically, different methods have been used to screen scoliosis. The most common method is the Adams test. In this test, the patient is asked to lean forward, and the examiner evaluates the symmetry of the trunk and the presence of dorsal or lumbar humps. During this test, an instrument known as a Scoliometer is usually used to measure the angle of trunk rotation (ATR) (Figure 1). Depending on the result, if it is less than 5°, no additional tests are required; if it is between 5° and 10°, a complete spine X-ray is usually performed. On the other hand, if the measurement is greater than 10°, an X-ray is always performed to rule out a possible scoliosis. Therefore, other techniques are being implemented in AIS screening, such as the Moiré topography. It uses a device that projects lines on the patient’s back. These are distorted according to the curvature of the body, creating a three-dimensional map that can be analyzed to detect asymmetries [4].

It is important to perform an adequate differential diagnosis of AIS with a scoliotic attitude. This latter is defined as a deviation of the rachis in the coronal plane with a possible appearance of humps in the Adams test; however, the ATR is less than 10° with a Scoliometer. In addition, a scoliotic attitude does not imply rotation of the vertebral bodies in radiographic tests, and the Cobb angle is usually less than 10°. In short, scoliotic attitudes are usually postural and not pathological [6]. For the definitive diagnosis of AIS, complementary tests such as telemetry in anteroposterior and lateral projection are usually requested. In this test, the measurement of the Cobb angle is important. In this way, scoliosis is considered when the Cobb angle is greater than 10°, and curvatures < 10° are considered normal. In addition, the Cobb angle is useful in determining the severity of curvature deviation [4,6,7].

The evaluation of bone maturity is a relevant prognostic factor in AIS. The Risser scale is based on the evaluation of the ossification of the iliac crest observed in the radiograph. This scale is classified into six grades ranging from grade 0 to 5; a low grade indicates less bone maturation and, consequently, a greater potential for growth and risk of progression of the curvature. However, a high grade indicates less growth and, therefore, less possibility of curvature progression (Figure 2). For these reasons, the Risser scale is a fundamental tool in therapeutic decisions about the optimal time to initiate treatment [8].

In addition, other factors associated with an increased risk of progression of the deformity are as follows: (1) a Cobb angle greater than 25°; (2) an age at diagnosis less than 13 years; (3) a history of AIS in immediate family members; (4) an accelerated growth velocity; (5) the presence of thoracic curves or double curves; and (6) a high apical rotation and low flexibility of curvature [7,9].

The clinical manifestations of AIS are variable. Mild curves are sometimes detected incidentally on examination after noticing certain signs such as asymmetry in the height of the shoulders, hips, or a truncal prominence. In contrast, more severe curves may be associated with low back pain, aesthetic deformities, and psychosocial anxiety. Therefore, an early diagnostic-therapeutic approach is important to avoid further progression of skeletal deformity and, consequently, further respiratory and cardiovascular complications [1,10]. In this sense, the vertebral rotation leads to deformities in the rib cage (Figure 3) and the appearance of dyspnea symptoms. In addition, this skeletal deformity and thoracic rigidity may cause a restrictive respiratory disorder with a reduction in forced vital capacity (FVC) and total lung volume (TPLV) in Spirometry. Other factors include a greater Cobb angle, an apical vertebral rotation, and a rib hump affecting lung expansion [11,12].

This entity has a relevant impact on long-term exercise tolerance. Regular monitoring of pulmonary function in these patients is recommended to prevent progressive respiratory deterioration. In this way, maintaining adequate spinal flexibility is crucial to preserving respiratory function [11,12]. Sometimes, these patients may present pulmonary hypertension secondary to pulmonary restriction, and it may contribute to the development of heart failure [14]. In addition, this entity can have an impact on mental health and health-related quality of life. These patients have a high prevalence of depressive and anxious symptoms related to aesthetic concerns and functional limitations. These subjective perceptions can be quantified in a standardized way using the SRS-22 questionnaire (Scoliosis Research Society-22). Therefore, it is important to emphasize the importance of carrying out not only a biomechanical correction, but also psychological and social support strategies to improve the health-related quality of life of these patients [15].

The therapeutic options available to patients with AIS range from conservative management based on observation, specific exercises, and the use of orthopedic braces to surgical treatment. The selection of the type of treatment depends on certain risk factors such as skeletal maturity, magnitude, and progression of the curvature, among others. Surgical treatment is usually indicated in patients with a curvature deviation greater than 40–45° according to the Cobb angle.

If the curvature is less than 20–25° according to the Cobb angle, observation with serial radiographs and exercise is usually recommended. On the other hand, if the Cobb angle is between 20–25° and 40–45°, conservative treatment is advised with the use of orthosis alone or associated with specific exercises to strengthen the dorsal lumbar musculature in these patients [4].

In recent years, exercise-based treatment has become relevant as a non-invasive treatment without side effects in the management of patients with AIS. Different specific exercise methods have been developed in order to correct structural deformity, improve postural control, and optimize musculoskeletal function. Some of them are the Schroth method, Lyon method, Scientific Exercise Approach for Scoliosis (SEAS), Barcelona School of Physical Therapy for Scoliosis (BSPTS), Dobomed method, Side-Shift method, functional independent treatment for scoliosis (FITS), core stabilization exercises, and exercises based on proprioceptive neuromuscular facilitation (PNF) [16].

In particular, the Schroth method was developed by Katharina Schroth in the early 20th century. However, various techniques have appeared, including Schroth Best Practice, Schroth 3D Treatment, Schroth General, and the BSPTS approach. The Schroth method is based on the three-dimensional correction of each patient’s specific curvature pattern through a combination of sensorimotor, postural, and corrective breathing exercises. Exteroceptive and proprioceptive stimulations aim to correct scoliotic posture through isometric exercises and strengthening asymmetrical muscles while maintaining a specific breathing pattern. A fundamental element of this approach is self-correction, defined as the patient’s ability to reduce the spinal deformity by actively realigning the spine in all three planes of space. In addition, it is recommended that patients integrate and maintain the corrected posture in their activities of daily living in order to maximize the benefits of treatment [17].

The modification of the breathing pattern is achieved with a technique called rotational breathing. This consists of taking deep, controlled breaths, directing the inhaled air towards the concave areas of the chest, thereby causing the contraction of the muscles in the convex area and mobilizing the ribs. The goal is to expand the areas of the rib cage that are compressed due to scoliosis. This type of breathing helps improve the symmetry of the rib cage and reduce trunk rotation (Figure 4) [17].

Several studies have shown that non-surgical methods, including the Schroth method, have positive effects in the treatment of patients with AIS. This conservative method has gained increasing relevance, avoiding the risks associated with surgical interventions [19]. For this reason, the main objective of this review was to analyze the effectiveness of the Schroth method applied either alone or in combination with other conservative therapies on variables such as the Cobb angle, angle of trunk rotation (ATR), pulmonary function, and quality of life in patients with AIS.

## 2. Materials and Methods

A comprehensive review was conducted using the PubMed electronic database. We applied following inclusion criteria to ensure the relevance, quality, and consistency of the studies: (1) studies published between 2020 to May 2025; (2) studies with greater scientific evidence (randomized controlled trials, systematic review and meta-analysis); (3) studies focus on Schroth method application during the most recent decade; (4) articles published in English; and (5) studies with full text access.

The literature search was conducted between September 2024 and March 2025. We included different combinations of the terms and MeSH terms “adolescent”, “idiopathic”, “scoliosis”, and “Schroth” connected with the Boolean operator “AND”.

The exclusion criteria were as follows: (1) studies published outside the range of 2020 to 2025; (2) studies with low scientific evidence (letter to authors, editorials, technical notes, case reports, case series and case reports); (3) studies not focused on the Schroth method; (4) articles published in a language other than English; and (5) studies without full text access.

The following data were extracted: (1) title; (2) authors; (3) publication year; (4) design; (5) characteristics of study participants (Cobb angle, Risser scale, curvature location, other orthopedic treatments); (6) intervention; and (7) outcomes.

Database bias and language bias have been considered, as the search was only conducted in the PubMed database, and the texts selected were in English.

To conduct this narrative review of the literature and a qualitative analysis of the studies obtained, we followed the recommendations of the SANRA guidelines, and their six quality criteria (justify the topic; state the objectives; describe the literature; present the evidence found; provide adequate discussion; and highlight the relevance of the topic).

## 3. Results

The initial bibliographic search in the PubMed database using the search criteria mentioned showed an amount of 82 records. Of these studies, 44 articles were excluded due to temporal and methodological criteria: 24 because they were not within the established date range (2020–2025) and 20 articles because full access to the text was not available, which is an essential requirement for critical evaluation.

After this initial screening, 38 studies were considered potentially relevant. However, 24 additional articles were excluded for not meeting the minimum scientific evidence criteria. Subsequently, 14 articles were subjected to a full review. One of them was discarded because it had been retracted from the corresponding scientific journal, leaving a final selection of 13 articles that met all the inclusion criteria established for this narrative review.

Of the studies included, five were RCTs and eight were systematic reviews or meta-analyses, which analyzed the effectiveness of the Schroth method in relevant clinical variables such as the Cobb angle, angle of trunk rotation (ATR), lung function, and health-related quality of life.

Figure 5 shows the flowchart and stages of the bibliographic search process for the texts analyzed. Likewise, Table 1 presents the data from these studies, classified according to the author and year of publication.

## 4. Discussion

The aim of this narrative review was to analyze the scientific evidence available in the literature on the effectiveness of the Schroth method in terms of reducing the Cobb angle and ART, and improving lung function, health-related quality of life, and postural control in patients with scoliosis. Similarly, the objective of this study was to review the available evidence comparing this method with other conservative approaches, as well as to identify its main limitations in clinical application and, therefore, to establish recommendations for future research. In this regard, based on the results of the studies reviewed, it is possible to carry out a rigorous analysis of the therapeutic benefits and main limitations of the Schroth method in the treatment of AIS.

As noted above, one of the purposes of our study was to evaluate the effect of the Schroth method on the Cobb angle. In this regard, numerous studies have evaluated the effectiveness of these exercises in reducing this parameter, obtaining statistically significant results and, therefore, proposing this treatment as an effective therapeutic option.

Several RCTs have directly compared the Schroth method to other specific exercises for AIS, including those by Kocaman, H., et al. [20] and Mohamed, R.A., et al. [21], who compared the Schroth method with core stabilization exercises and PNF exercises, respectively. In both cases, the group treated with Schroth showed a reduction in the Cobb angle that was not only statistically significant (*p* < 0.05 in both) but also exceeded the clinically relevant threshold of 5° in mild curves and Risser ≤ 3. It is important to note that this study did not evaluate the effect of Schroth exercises as a single treatment, but rather as a supplement to standard treatment.

Other studies, such as those by Ceballos-Laita, L., et al. [23], Chen, C., et al. [28], and Kyrkousis, A., et al. [30], found greater effectiveness in improving deformity in patients with mild curves and low Risser stages (Risser 0–3), with statistically significant results (*p* < 0.05). These findings reaffirm the importance of initiating treatment in the early stages of AIS.

In contrast, some of the meta-analyses analyzed, such as those by Ceballos-Laita, et al. [23], Chen, Y., et al. [25], and Wang, Z., et al. [31], found certain relevant nuances when comparing the Schroth method to other conservative treatment programs. Ceballos-Laita et al. concluded that the Schroth method alone is effective in reducing the Cobb angle and the trunk rotation angle, and improving quality of life in the short term, compared to no intervention or other conservative treatments. However, the improvement in the Cobb angle did not exceed the minimum clinically significant difference. This improvement was 3.18°. On the other hand, Wang, Z., et al. [31] ranked SEAS superior to Schroth, obtaining better results in terms of improvement in the Cobb angle.

On the other hand, certain studies have compared the Schroth method to standard treatment (observation and/or orthopedic brace) in patients with AIS, and have even evaluated the effectiveness of combining both treatments. Among them, the study by Kyrkousis, A., et al. [30] demonstrated that combining the Schroth method with the use of a brace was effective in reducing the average Cobb angle by 5.63°, which was significantly greater than that observed with the use of a brace alone (*p* < 0.001). However, long-term follow-up after discontinuing the Schroth program found that the corrective effects on the spinal curve progressively attenuated, especially in patients with lower compliance with the home program.

Other studies, such as those by Fahim, T., et al. [22], concluded similar results that the combination of specific exercises and braces resulted in better outcomes than either intervention alone. In this regard, Schroth exercises provided additional benefits to the structural and functional components that the brace did not address, such as improved motor control, postural symmetry, and aesthetic perception. In addition, it was noted that prolonged use of the brace could lead to adverse effects such as muscle atrophy and decreased thoracic mobility, reinforcing the need to integrate programs such as Schroth that enhance muscle and respiratory function during treatment. However, Khaledi, A., et al. [27] concluded no statistically significant differences between Schroth exercises versus core stabilization exercises in terms of Cobb angle reduction (*p* > 0.05).

Dimitrijević, V., et al. [19] identified that the Schroth method showed the greatest effect size in reducing the Cobb angle and improvement of health-related quality of life (*p* < 0.05). However, no significant differences were found in ATR (*p* = 0.06); the differences were not statistically significant when compared directly with other exercises. Similarly, Chen Y et al. [25] concluded in their systematic review that some exercise programs, such as the core-based approach or SEAS, could outperform the Schroth method in reducing the Cobb angle, although they acknowledged that the evidence was inconclusive and could be justified by a low representation of studies that included the Schroth intervention (only 1 of 12 trials included).

On the other hand, another variable evaluated in the published studies was ATR. This is one of the most relevant parameters on which the Schroth method acts. In most of the studies analyzed, the improvement in ATR reduction was statistically significant.

Among the studies that assess the impact of the Schroth method on this parameter are the trials by Kocaman, H., et al. [20] and Mohamed, R.A., et al. [21], which showed similar results in significant ATR reduction in the group treated with Schroth compared to other exercises (*p* < 0.001), although in the case of lumbar rotation, Schroth exercises did not show significant differences compared to core exercises. Complementarily, Zhang, Y., et al. [29] demonstrated that the combination of pelvic rotation correction with Schroth exercises was more effective than Schroth exercises alone in improving spinal and pelvic deformities. No significant differences in the ATR and Cobb angle were found (*p* > 0.05).

Furthermore, Kyrkousis, A., et al. [30] observed a substantial improvement in ATR after combining the Schroth method with the use of a brace, an effect that was maintained in the long term, 6 months after the intervention had ceased (*p* < 0.001).

This corrective effect on ATR has been directly linked to an improvement in aesthetic deformity, one of the most relevant aspects for adolescent patients. Several studies, such as those by Mohamed, N., et al. [23] and Kyrkousis, A., et al. [26], used postural analysis methods or subjective scales such as the Trunk Appearance Perception Scale and reported a significant reduction in trunk asymmetry. Similarly, Ceballos-Laita, et al. [30] confirmed that the Schroth method was associated with a significant reduction in ATR and improved aesthetic perception in patients with Risser stage 0–3. These aesthetic improvements not only affect body self-perception but can also positively influence therapeutic adherence and the patient’s health-related quality of life.

Along these lines, another consistent finding in this review is that the Schroth method improves the health-related quality of life of adolescents with AIS, as a result of not only structural, but also functional and emotional changes. Studies comparing the Schroth method to other exercises showed improvements in quality of life measured with the SRS-22, with superior results in the Schroth group [20,21,27]. Similar results were observed in studies comparing Schroth associated with conservative treatment versus standard treatment alone, especially in the domains of body image and physical function [23,24,26,30,31]. However, one limitation of some included studies was the way in which health-related quality of life was measured using scales such as the SRS22 and/or SRS-22r questionnaire. In this sense, the SRS-22r questionnaire was designed for evaluating health-related quality of life in children and adolescents with scoliosis who have undergone surgical treatment, not non-surgical treatment. Nevertheless, there are questionnaires that are more appropriate for assessing health-related quality of life for conservative treatments, such as EQ-5D-Y (EuroQol 5 Dimensions Youth Version) [32] and BrQ (Brace Questionnaire) [33], which should be considered in future studies.

One of the least studied aspects in the literature, which this review sought to investigate, was the impact of the Schroth method on lung function. The reviews by Khaledi, A., et al. [24] and Scheirber, S., et al. [27] pointed out the lack of evaluation of lung function in the studies reviewed, despite the fact that one of the fundamental physiological principles of the Schroth method is three-dimensional breathing and chest expansion.

In this regard, only the meta-analysis by Dimitrijević, V., et al. [19] studied the impact of the Schroth method on lung function, finding a significant improvement in FEV1, suggesting that the Schroth method could have real benefits on respiratory capacity. However, no significant changes in FVC were observed. This result highlights the need to design studies that include a systematic assessment of lung function in patients with AIS, especially in thoracic scoliosis, where respiratory compromise is common.

However, after reviewing these studies, it is important to consider a number of common methodological limitations that affect the robustness and generalizability of the results obtained.

First, it should be noted that few studies have conducted long-term follow-up after the completion of treatment with Schroth method-based exercises, making it difficult to assess whether the benefits in the Cobb angle or ATR reduction are maintained after treatment completion, especially in patients who have not completed skeletal maturation. Specifically, Schreiber, S., et al. pointed out that no study achieved a clinically relevant reduction (≥5°) in the Cobb angle in the long term due to the short follow-up periods and methodological limitations of the included trials [24].

Likewise, the Schroth method requires active patient participation to achieve significant clinical effects. However, few reviewed studies evaluate therapeutic adherence to the exercises. Nevertheless, the study by Mohamed, N., et al. [26] considered this aspect and demonstrated that patients with high adherence to the Schroth program achieved much more pronounced improvements in objective postural alignment, subjective body perception, and quality of life. Likewise, Khaledi, A., et al. [27] pointed out the importance of adequate frequency, supervision, and duration of treatment to obtain better and sustained results over time with the Schroth method.

Another limitation of the studies analyzed is the lack of subgroup analysis according to curve type, severity, or skeletal maturity. This omission limits the possibility of identifying which adolescent profiles would benefit most from the application of the Schroth method or whether its effectiveness varies depending on these clinical variables.

In addition, several of the studies evaluated showed great heterogeneity in their design (variables, follow-up time, and frequency of intervention, among others), which makes it difficult to directly compare the studies and the results obtained. However, the meta-analysis by Ceballos, L., et al. [23] positioned the Schroth method as superior to other exercises (Pilates, core stabilization, and PNF), with no heterogeneity among the included studies.

On the other hand, although one of the theoretical pillars of the Schroth method is three-dimensional respiratory work, few studies have specifically evaluated lung function. Only a small number of studies included objective measurements of respiratory capacity (FEV1, FVC). This represents a significant limitation in understanding the benefits of the Schroth method in patients with thoracic scoliosis, where the respiratory impact can be significant.

Finally, Khaledi, A., et al. [27] highlight the ethical limitations of applying the Schroth method exclusively to patients who are actually indicated for bracing, which conditions the design of clinical trials and limits the ability to draw accurate conclusions about its isolated effectiveness in moderate to severe curves. In addition to these limitations, most of the studies reviewed are RCTs and refer to the efficacy of the Schroth method under optimal experimental conditions, which makes it difficult to correctly extrapolate the results to routine clinical practice to assess its effectiveness.

Consequently, all these limitations highlight the need for future research of higher quality with designs that are more rigorous, standardized homogeneous protocols, and prolonged follow-up, which will allow for the conclusive validation of the Schroth method as a conservative treatment for adolescent idiopathic scoliosis.

In this regard, the strengths of the Schroth method lie in its precise methodological structure, its three-dimensional approach, and the integration of the respiratory component. However, its main clinical weakness is the need for highly trained personnel and intensive supervision, which limits its applicability in settings with fewer resources.

## 5. Conclusions

The evidence reviewed in this study supports the Schroth method as an effective conservative intervention in the treatment of AIS, with documented benefits in reducing the Cobb angle, ATR, aesthetic perception, and quality of life. These effects are particularly evident in patients with mild curves and in the early stages of skeletal growth, highlighting the importance of early intervention. While several studies have demonstrated the superiority of Schroth over other specific exercise programs and over standard treatment alone (observation and/or brace), the magnitude of the effect does not always reach clinical significance in all contexts.

However, this review also highlights key limitations in the current literature. The effectiveness of the Schroth method appears to be strongly influenced by patient adherence, the intensity and duration of the program, and professional supervision, factors that are not always considered in the available studies; however, they are relevant and should be considered in future studies. Furthermore, the lack of long-term follow-up prevents the sustainability of the benefits obtained from being determined, and the heterogeneity in protocols and measurement tools makes it difficult to directly compare interventions, obtain accurate conclusions, and extrapolate results. In addition, it could be relevant that future studies assess lung function because one of the fundamental physiological principles of the Schroth method is three-dimensional breathing and chest expansion.

Overall, the findings of this review demonstrate the effectiveness of the Schroth method and its potential inclusion in therapeutic programs for adolescents with AIS alone or in addition to bracing if indicated. However, it is essential to conduct studies of higher methodological quality, with standardized protocols, analysis by clinical subgroups, and prolonged evaluation over time, to consolidate its effectiveness and define its indications more precisely as a conservative treatment in the multidisciplinary approach to scoliosis.

## Figures and Tables

**Figure 1 healthcare-13-02631-f001:**
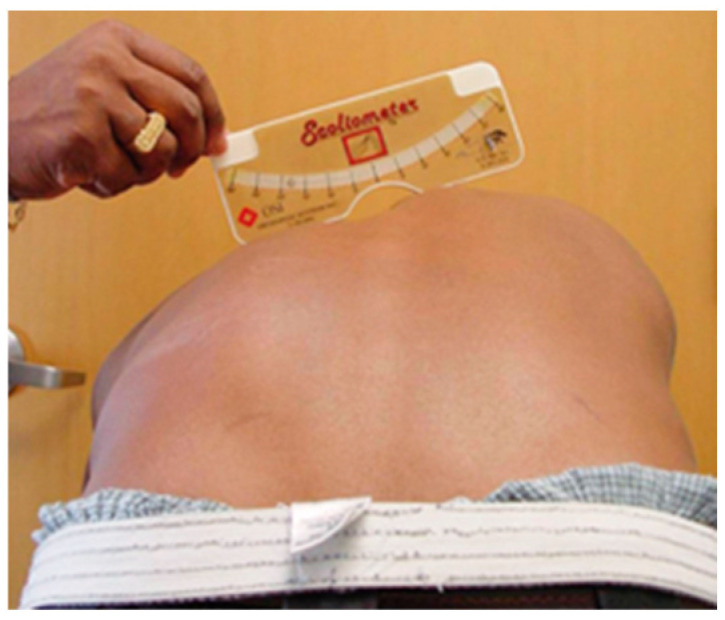
Adams test and ATR measurement with Scoliometer. Source: Álvarez García de Quesada LI, Núñez Giralda A. Idiopathic scoliosis. *Pediatr Aten. Primaria*. 2011;13(49):135–46 [5].

**Figure 2 healthcare-13-02631-f002:**
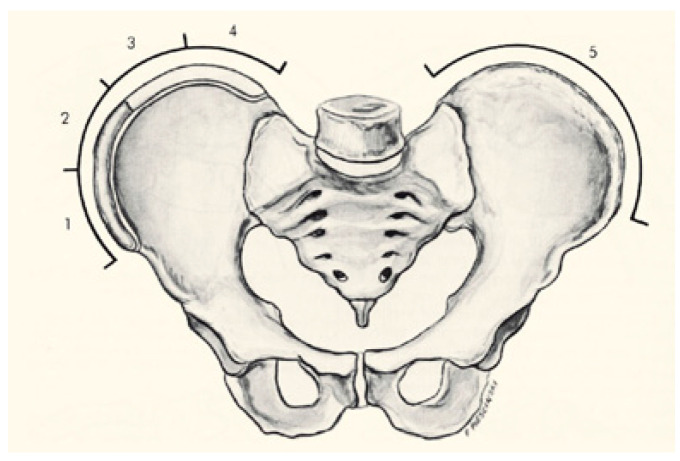
Risser scale. Source: Álvarez García de Quesada LI, Núñez Giralda A. Idiopathic scoliosis. *Pediatr Aten. Primaria*. 2011;13(49):135–46 [5].

**Figure 3 healthcare-13-02631-f003:**
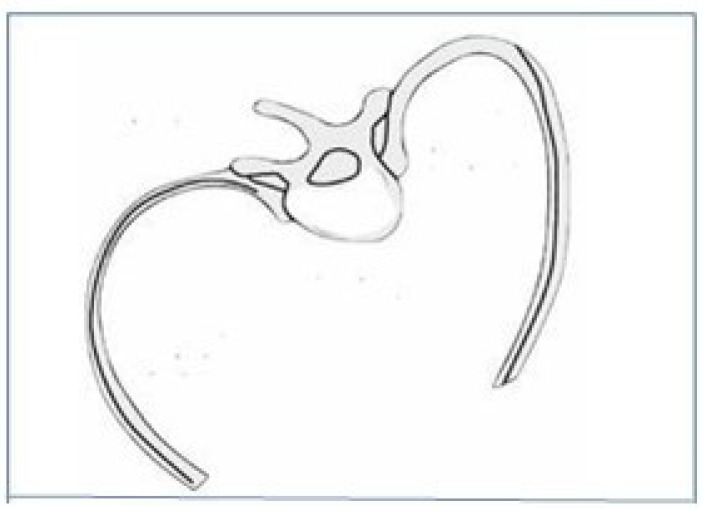
Diagram of thoracic deformity caused by vertebral rotation. It explains the elevation of the convexity, which determines the asymmetry in the Adams test. Source: Pantoja TS, Chamorro LM. Scoliosis in children and adolescents. *Rev médica Clin Las Condes*. 2015;26(1):99–108 [13].

**Figure 4 healthcare-13-02631-f004:**
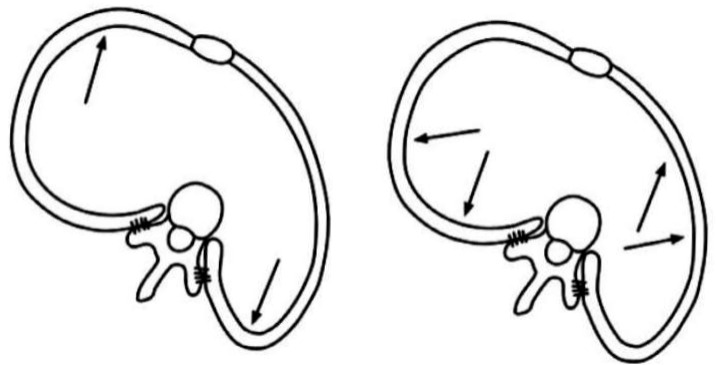
Rotational breathing. Source: Moramarco K, Borysov M. A modern historical perspective on Schroth scoliosis rehabilitation and bracing techniques for idiopathic scoliosis. *Open Orthop J*. 2017;11:1452–1465 [18].

**Figure 5 healthcare-13-02631-f005:**
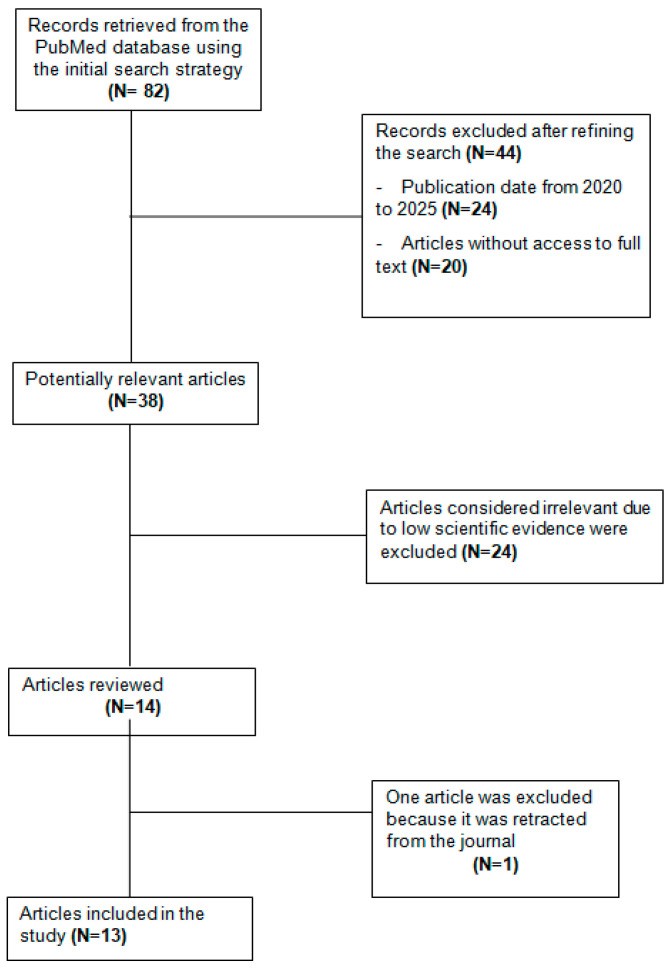
Flowchart.

**Table 1 healthcare-13-02631-t001:** Results of selected studies in narrative review.

STUDY	OBJECTIVE	METHODS	CONCLUSION
**Kocaman H, et al., 2021 [20]**	To evaluate effectiveness of Schroth method versus Core stabilization exercises.	RCT. Cobb angle (10–25°). Group Schroth and Group of Core exercises. Variables: Cobb angle, ATR, esthetic, joint, muscle balance, and health-related quality of life.	Schroth exercises were effective in improving Cobb angle, ATR, esthetic, joint and muscle balance, and health-related quality of life (*p* < 0.05).
**Mohamed RA, et al., 2021 [21]**	To assess the effectiveness of Schroth versus PFN exercises.	RCT. Cobb angle (10–25°). Risser ≤ 3 Group Schroth and Group of PFN. Variables: Cobb angle, ATR, plantar pressure, and pulmonary capacity.	Schroth exercises were more effective in reducing Cobb angle, ATR, plantar pressure, and pulmonary capacity (*p* < 0.05).
**Fahim T, et al., 2022 [22]**	To analyze the effectiveness of different physiotherapy interventions.	Systematic review. Schroth method, SEAS and core stabilization exercises, orthosis, and combination of Cobb angle.	The Schroth method showed a greater reduction in Cobb angle and this reduction was greater when combined with orthosis (*p* < 0.05).
**Ceballos-Laita L, et al., 2023 [23]**	To study the effectiveness of Schroth method versus other conservative treatments or no intervention.	Systematic review and meta-analysis Schroth method alone versus other conservative treatments or no intervention on Cobb angle, health-related quality of life, and ATR.	The Schroth method alone was effective in the short term in reducing Cobb angle, ATR, and improving quality of life (*p* < 0.05).
**Schreiber S, et al., 2023 [24]**	To review the scientific evidence of Schroth method.	Systematic review. Cobb angle (10–45°). The quality of published studies was evaluated.	Improvement of quality of life but no significant reduction in Cobb angle (*p* < 0.05). Methodological quality limits their validity.
**Chen Y, et al., 2023 [25]**	To evaluate the effect of exercises versus conventional therapies on Cobb angle.	Systematic review and meta-analysis. Schroth, core stabilization, yoga, and suspension exercises versus conventional therapies.	The exercises were more effective in reducing Cobb angle than conventional therapies, without differences between different exercises (*p* < 0.001).
**Mohamed N, et al., 2024 [26]**	To compare Schroth method associated with standard treatment versus standard treatment alone.	RCT. Cobb angle (10–45°), Risser ≤ 3 Group Schroth + standard treatment and Group standard treatment alone on ATR	The combination of Schroth exercises to standard treatment showed a reduction in ATR (*p* < 0.05).
**Khaledi A, et al., 2024 [27]**	To evaluate the available evidence on the effectiveness of Schroth exercises versus core stabilization exercises.	Systematic review. Schroth versus core stabilization on Cobb angle.	No statistically significant differences were found between the two methods in terms of Cobb angle reduction (*p* > 0.05).
**Chen C, et al., 2024 [28]**	To assess the evidence on effectiveness of Schroth exercises versus conventional treatment.	Systematic review and meta-analysis. Variables: Cobb angle, ATR, muscle strength, and health-related quality of life	The Schroth method was more effective in reducing Cobb angle, ATR, increasing muscle strength, and improving health-related quality of life (*p* < 0.001).
**Zhang Y, et al., 2024 [29]**	To evaluate the effectiveness of PNF-based pelvic rotation correction combined with Schroth exercises versus Schroth exercises alone.	RCT. Group Schroth exercises + PNF and Group Schroth exercises alone. Variables: pelvic asymmetry index, Cobb angle, ATR, and health-related quality of life.	The combination of pelvic rotation correction with Schroth exercises was more effective than Schroth exercises alone in improving spinal and pelvic deformities. No significant differences in ATR and Cobb angle (*p* > 0.05).
**Kyrkousis A, et al., 2024 [30]**	To evaluate the effectiveness of Schroth exercises and orthoses versus orthoses alone.	RCT. Group Schroth exercises + brace and Group brace. Variables: Cobb angle, ATR, and health-related quality of life.	Schroth exercises and orthopedic treatment demonstrated a reduction in Cobb angle, ATR, and health-related quality of life (*p* < 0.001).
**Wang Z, et al., 2024 [31]**	To evaluate the evidence in the literature on effectiveness of different exercises on spinal deformity and health-related quality of life.	Systematic review and meta-analysis. Specific exercises compared to routine care, bracing, and general exercises. Variables: Cobb angle, ATR, and health-related quality of life.	Schroth method was effective in the short and long term in reducing Cobb angle and improving quality of life. Active self-correction showed the best short-term results (*p* < 0.05).
**Dimitrijević V, et al., 2024 [19]**	To review the evidence of the effect of different exercises on AIS.	Systematic review and meta-analysis. The effect of different types of exercises was evaluated. Variables: Cobb angle, ATR, lung function, and health-related quality of life.	Schroth method had positive effect on reduction in Cobb angle, and improvement of health-related quality of life and FEV1 (*p* < 0.05). No significant differences were found in ATR and FVC (*p* = 0.06).

RCT: Randomized Controlled Trial; ATR: Angle of Trunk Rotation; PNF: Proprioceptive Neuromuscular Facilitation.

## Data Availability

No new data were created or analyzed in this study.

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
