# Peer review of "The Effects of the Schroth Method on the Cobb Angle, Angle of Trunk Rotation, Pulmonary Function, and Health-Related Quality of Life in Adolescent Idiopathic Scoliosis: A Narrative Review"

_healthcare, 2025, doi:10.3390/healthcare13202631_

Round 1

Reviewer 1 Report

Comments and Suggestions for Authors

Search limited to PubMed only → creates database bias. Authors themselves acknowledge this but it weakens validity.

No PRISMA-style flow diagram details beyond simple numbers.

Results lack quantitative synthesis (effect sizes, confidence intervals). Mostly narrative and descriptive.

Some tables poorly formatted; inconsistent alignment.

Comments on the Quality of English Language

Several grammatical errors: e.g., “affects to female sex” (should be “affects females”), “pretend a correction” (should be “aim to correct”).

Redundant phrasing in places (“this entity is the form most frequent”).

Author Response

Dear Reviewer 1,

We thank you for your comments. We will try to explain our reply below. We carried out the following modifications:

  • You are right that searching only the PubMed database weakens the validity and the generalization of the findings of this study and it creates the database bias that we acknowledge. However, in our case, we chose a single database, PubMed, because it is the most widely used in our service and has comprehensive information in the field of biomedicine. In addition, the objective of this review was to conduct a critical and conceptual analysis of the effectiveness of the Schroth method in adolescent idiopathic scoliosis in a short period of time (September 2024 and March 2025) and PubMed has unrestricted access to most of its articles, which makes it easy to read the full text.
  • There are no PRISMA-style details in the flow diagram because this study is neither a systematic review nor a meta-analysis. Narrative reviews do not require following standardized search and selection processes as rigid as those used in systematic reviews or meta-analysis. To conduct this narrative review, we followed the most appropriate guide for this type of review, SANRA guidelines, and their six quality criteria (justify the topic; state the objectives; describe the literature; present the evidence found; provide adequate discussion; and, highlight the relevance of the topic). This information has been added in red letter at the end of the materials and methods section of the manuscript.
  • You are right about presenting results in narrative and descriptive form. It is because a narrative review is characterized by a qualitative and descriptive analysis and their purpose is to synthesize, organize, and discuss the evidence, rather than integrate it statistically. Quantitative analysis with effect sizes and confidence intervals is more characteristic of meta-analyses, which are a higher level of quantitative synthesis within systematic reviews.
  • We have modified the format of the tables and have tried to align them as best as possible, as well as remove irrelevant information.
  • The manuscript was reviewed by a native English teacher, and expressions and grammatical errors were modified in red letter. Thank you for pointing out these grammatical errors; we have modified them in the manuscript. We have replaced “affects to female sex” with “affects females” and “pretend a correction” with “aim to correct.”
  • We have tried to avoid excessive redundancy in the sentences in the manuscript.

Reviewer 2 Report

Comments and Suggestions for Authors

healthcare-3859320-peer-review-v1

Thank you for the opportunity to review the manuscript entitled “The effectiveness of Schroth method in Cobb angle, angle of 2 trunk rotation, pulmonary function and health-related quality 3 of life in adolescent idiopathic scoliosis: a narrative review.”

The reviewed article provides a thorough narrative review of the Schroth method in adolescent idiopathic scoliosis, examining its effects on clinical outcomes such as Cobb angle, trunk rotation , pulmonary function, and quality of life. A key strength lies in its systematic selection of evidence, focusing on high-quality studies—RCTs, systematic reviews, and meta-analyses published between 2020 and 2025—which enhances the credibility and relevance of the findings. The review also offers detailed comparisons with other conservative treatments, supported by clear tables and figures (e.g., study selection flowchart, Adams test, Risser scale), thereby situating the method within clinical practice.

The discussion highlights the Schroth method’s benefits, particularly in reducing Cobb angle and trunk rotation in patients with mild scoliosis and early skeletal growth. The authors emphasize its three-dimensional corrective exercises combined with rotational breathing, noting improvements not only in spinal structure but also in functional, psychosocial, and aesthetic outcomes. They further address the added value of combining Schroth therapy with bracing, which helps reduce brace-related muscle atrophy and supports respiratory function.

The review has several methodological and interpretive limitations. It relied solely on PubMed and English-language studies, creating risks of publication and language bias while excluding relevant databases. Although it claimed to review high-level evidence (RCTs, systematic reviews, meta-analyses), the final selection mixed study designs without a clear hierarchy, blurring its rigor. The narrative review format is inconsistently presented as systematic, potentially misleading readers.

Significant heterogeneity across included trials—sample sizes, protocols, follow-up, and outcomes—was not addressed through subgroup analyses (e.g., curve severity, skeletal maturity, brace use). Effectiveness claims often downplayed contradictory evidence from alternative methods and overstated improvements that did not reach clinically meaningful thresholds. Short follow-up periods further limited conclusions on long-term benefits.

Respiratory outcomes, central to Schroth therapy, were scarcely assessed, and the evidence presented was weak. Critical factors such as adherence, supervision intensity, therapist availability, and accessibility in resource-limited settings were only superficially discussed. Finally, placing some patients who required bracing into Schroth-only groups raises ethical concerns and limits generalizability.

Author Response

Dear Reviewer 2,

We thank you for your comments. Thank you for considering the strengths and weaknesses of this manuscript. We will try to explain our reply below. We carried out the following modifications:

  • No other databases were consulted because recent articles from the last years with high-scientific evidence were found in PubMed. We know that this could weaken the validity and generalization of the findings of this study and it creates the database bias. However, in our case, we chose a single database, PubMed, because it is the most widely used in our service and has comprehensive information in the field of biomedicine. In addition, the objective of this review was to conduct a qualitative analysis of the effectiveness of the Schroth method in adolescent idiopathic scoliosis in a short period of time ((September 2024 and March 2025) and PubMed has unrestricted access to most of its articles, which makes it easy to read the full text. You are right that we considered articles published in English only and it can lead to language bias, but most of the articles were published in English and they had a high level of scientific evidence. This information has been added in red letter at the end of the materials and methods section of the manuscript.
  • In this narrative review, studies with the highest level of scientific evidence (clinical trials, systematic reviews, and meta-analyses) were selected, and other studies with a lower level of evidence were excluded. The final selection of study designs was presented in hierarchical order according to year of publication, from oldest to most recent. This narrative review followed the SANRA guidelines, and their six quality criteria (justify the topic; state the objectives; describe the literature; present the evidence found; provide adequate discussion; and, highlight the relevance of the topic). This information has been added in red letter at the end of the materials and methods section of the manuscript.
  • Narrative reviews do not require following standardized search and selection processes as rigid as those used in systematic reviews or meta-analysis.
  • The significant heterogeneity among the included trials (sample sizes, protocols, follow-up, and outcomes) was addressed by subgroup analyses. We compared mild curves with a Cobb angle between 10º and 25º and moderate curves with a Cobb angle between 10º and 45º. Skeletal maturity was also considered, comparing only patients with high risk of curve progression with Risser stage 0-3 and whether or not they wore a brace. This information has been added and remarked in red letter in tables and discussion section of the manuscript.
  • The conclusions on the comparison of the Schroth method with other therapeutic alternatives took into account the benefits achieved with the latter and their relevance. However, we sought above all to highlight the results obtained with the Schroth method because it is a conservative and innovative option and its complementary use with other treatments could be useful in clinical practice. In most cases, the improvements were clinically and statistically significant. This information has been added, clarified and remarked in red to the text, indicating which articles did not exceed the clinically and statistically significant difference.
  • It is true that short follow-up periods further limited conclusions on long-term benefits. For this reason, it is necessary studies with long follow-up periods. This information has been remarked in red letter in conclusion section of the manuscript.
  • It is indicated in second paragraph of the conclusions that although certain critical factors such as adherence, supervision intensity, therapist availability, and accessibility in resource-limited are not considered in studies analyzed, it is important to take into account and should be considered in future studies. This information has been remarked in red letter in conclusion section of the manuscript.
  • It is true that there is a lack of assessment of lung function in the studies reviewed. Scheiber et al., indicate that despite the fact that one of the fundamental physiological principles of the Schroth method is three-dimensional breathing and chest expansion. However, other studies such as Dimitrijević V et al. take into account changes in parameters as FEV1 and FVC in spirometry. This information has been remarked in red letter in tables and discussion section of the manuscript.
  • The Schroth method is a therapeutic option that could be useful as a treatment for mild and moderate curves with a high risk of progression. It is not intended to replace orthopedic treatment in cases where it is indicated, but rather to be a complementary treatment for moderate curves and a useful treatment option for mild curves.

Reviewer 3 Report

Comments and Suggestions for Authors

While this narrative review of the literature is well prepared and thorough, it is not a systematic review. The review was limited to a 5 year span of time, something for which was no rationale.  The authors identified 13 articles for inclusion (5 RCTs and 8 SRs w/ MA). I find now reason how a narrative review of the literature could add value to the literature when it is flush with higher quality studies on the exact same topic.

Comments on the Quality of English Language

English needs significant work throughout text. Several typos exist as well. 

Author Response

Dear Reviewer 3,

We thank you for your comments. Thank you for considering narrative review of the literature is well prepared and thorough. We will try to explain our reply below. We carried out the following modifications and considerations:

  • As you correctly point out, this work is a narrative review and not a systematic review, as indicated in the title of this manuscript.
  • The review covered the last five years in order to assess the most recent scientific evidence. It was also noted that very few articles had been published beyond the five-year period.
  • The objective of this review was to conduct a qualitative analysis of the effectiveness of the Schroth method in adolescent idiopathic scoliosis in a short period of time (September 2024 and March 2025). In this narrative review, studies with the highest level of scientific evidence (clinical trials, systematic reviews, and meta-analyses) were selected, and other studies with a lower level of evidence were excluded. The final selection of study designs was presented in hierarchical order according to year of publication, from oldest to most recent. This narrative review followed the SANRA guidelines, and their six quality criteria (justify the topic; state the objectives; describe the literature; present the evidence found; provide adequate discussion; and, highlight the relevance of the topic). This information has been added in red letter at the end of the materials and methods section of the manuscript.

We believe that conducting a narrative review including articles with high scientific evidence such as those included is relevant. This provides us information and conclusions about a topic that could be extrapolated to daily practice. The Schroth method is a conservative and non-invasive treatment that can complement orthopedic treatment in moderate curves to improve the health-related quality of life of patients with scoliosis and other associated comorbidities. In addition, it could be considered a treatment option in mild curves with high risk of progression. Although further studies are needed, this narrative review suggests that this could be a novel treatment and encourages further research in this area considering the limitations presented in this manuscript.

  • The manuscript was reviewed by a native English teacher, expressions and grammatical errors were modified in red letter in the manuscript.